# Use of generative AI for health among urban youth in Pakistan: A mixed-methods study

Ahsan Mashhood*, Aamna Ahmed, Inaya Khan, Maryam Hashim, Sara Baloch

Department of Social Development Policy, Habib University, Karachi, Pakistan

* ahsan.mashhood@ahss.habib.edu.pk

## Abstract

Generative AI (GAI) tools are increasingly used informally for health, yet evidence from low- and middle-income countries (LMICs) is limited. This study generates early evidence on such health systems from the fifth most populous country: Pakistan. We used a youth-led convergent mixed-methods design among digitally connected urban youth in Pakistan (survey N = 1240, 20 interviews). The primary outcome was any GAI use for health. We fitted multivariable logistic regression models and conducted reflexive thematic analysis. Overall, 69.0% of participants reported using GAI for health. Higher odds of use were observed among women (aOR = 1.57, 95% CI [1.17–2.11], p = 0.003) and youth reporting any mental or physical condition (aOR = 1.82, 95% CI [1.34–2.48], p < .001). Greater trust in AI strongly predicted use (per-level aOR = 4.21, 95% CI [2.98–6.01], p < .001). High confidence using AI (aOR = 1.81, 95% CI [1.11–3.07], p = 0.022), awareness of AI risks (aOR = 1.67, 95% CI [1.20–2.31], p = 0.002), and prior use of other (non-generative) digital health tools (aOR = 4.48, 95% CI [2.59–8.23], p < .001) were also associated with higher likelihood of use. Telemedicine use was significant though weaker in magnitude (aOR = 1.58, 95% CI [1.01–2.54], p = 0.049). Interviews highlighted three themes: (1) access and affordability driving first-line use; (2) emotional safety and informational support, especially for stigmatized concerns; (3) perceived empowerment in interpreting tests, organizing symptoms, and preparing for clinical visits. Given constrained, stigmatizing, and costly services, GAI may function as an adjunct step for health information and emotional support in Pakistan's health ecosystem.

## Author summary

Generative artificial intelligence (GAI) tools such as ChatGPT are increasingly part of young people's everyday lives, yet we know very little about how they are used for health in low- and middle-income countries. This mixed-methods, youth-led study explores how and why digitally connected urban youth in Pakistan use GAI for questions about their bodies, emotions, relationships, and daily health

**Data availability statement:** Both de-identified data and code are available on OSF https://osf.io/uk7ct/.

**Funding:** AM received the grant by Habib University https://habib.edu.pk/contact-us/ with the code HU-ERC-2025-AM1. The funders had no role in study design, data collection and analysis, decision to publish, or preparation of the manuscript.

**Competing interests:** The authors have declared that no competing interests exist.

concerns. We combined an online survey of 1240 young people with in-depth interviews with 20 routine users. Many participants report turning to GAI for quick, anonymous, and non-judgmental guidance, especially for sensitive topics they may feel unable to discuss with family or health professionals. At the same time, young people are aware that AI can be inaccurate or biased and often describe checking information across multiple sources. Our findings, which we frame as exploratory and hypothesis-generating, suggest that GAI is becoming an informal layer in the health information ecosystem for urban youth. Rather than ignoring this emerging trend, public health actors in Pakistan and similar settings may need to understand, monitor, and ethically engage with these emerging practices to reduce, rather than widen, existing health and digital inequalities.

## 1. Introduction

In recent years, generative artificial intelligence (GAI) tools such as ChatGPT, Gemini (Bard) and Claude have rapidly entered the everyday lives of digitally connected youth [1,2]. These tools offer an unprecedented form of real-time, anonymous, and conversational information delivery, which distinguishes them from earlier digital health technologies such as search engines, static health websites, and scripted rule-based chatbots. While recent debates around GAI have largely centered on misinformation, copyright, and productivity [3,4], surprisingly little is known about how these tools are being used informally for health-seeking, particularly in the Global South [5,6], where youth often face stigma, limited access to health services, and structural barriers to care [7,8,9,10,11].

In Pakistan, a country with a median age of 21 [12] and socio-cultural constraints around health disclosure [13,14], these tools may be forming an infrastructure of self-directed care by offering nonjudgmental, always-available alternatives to a relatively dysfunctional healthcare system [8,10]. A deluge of local social media content suggests that young people may be turning to GAI for advice related to anxiety, sexual health, acne, fitness, nutrition, and other concerns they may feel uncomfortable discussing with parents or doctors. However, this remains an understudied phenomenon. Little to no research efforts have comprehensively explored how young people in Pakistan (or the Global South) are using and appropriating unregulated GAI tools to meet their health needs, nor how individual, relational, or community factors interact to shape this behavior.

Existing research, mostly from the Global North, investigates specific chatbots/companion apps like Woebot [15] or Replika [16], rather than general-purpose GAI tools like ChatGPT that young people are already using (e.g., for education) or are increasingly familiar with [17], also see S1 Fig). To our knowledge, only one published study from the Global South [18] examines such uses, underscoring how nascent and underexplored this field remains in these contexts. Accordingly, this paper contributes: (i) early evidence from a large-sample LMIC estimate of youth GAI health use; (ii) a theory-linked account (socio-ecological) of who uses GAI and why;

(iii) a conceptual model on the emerging youth-AI health support pathway; (iv) policy, practice, and systems implications for integrating GAI in youth health, especially in the LMIC context.

## 1.1. Theoretical framework

To examine this emerging phenomenon, we draw on the socio-ecological model as our guiding theoretical framework, particularly its adaptation by Mansfield and colleagues [19]. This model conceptualizes health behaviors as the outcome of dynamic interactions between (1) individual (e.g., demographic), (2) relational (e.g., interpersonal) and, (3) community factors (Fig 1). These are critical to understanding the Pakistani context. Accordingly, extending this framework to the context of GAI health use, we consider how individual demographics (gender, socio-economic class, sexual orientation, pre-existing conditions etc.) intersect with interpersonal dynamics (e.g., perceived social support; friends and family), and community structures (e.g., healthcare access), to shape how and why young people turn to GAI for health information and potentially, decision-making. In doing so, we explore health-seeking via GAI as an emergent response, worth timely exploration in Pakistan's layered and often exclusionary health ecosystem [20].

By collecting primary data on digitally connected urban youth aged 18–30 in Pakistan, this study investigates the prevalence, predictors and perceptions associated with GAI use for health purposes. More simply put, we center youth and use socio-ecological theory to examine who is using GAI for health and why, to discuss implications for social policy, health literacy, and the ethical governance of GAI systems. In doing so, we contribute new empirical evidence and a conceptual model to this nascent subfield of digital public health, while also raising critical questions about the big picture, i.e., what

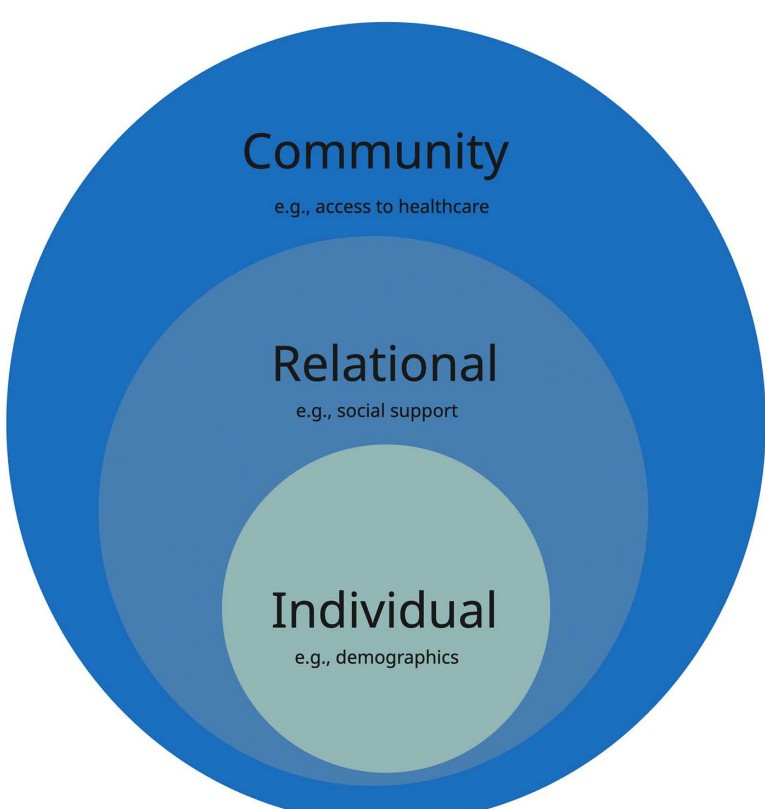

**Fig 1. Socio-ecological Model ([19],CDC, 2022).**

constitutes care in this new era of algorithmic therapeutic alliance and the consequent ethics of such potential 'task shifting' via GAI.

## 2. Methods

### 2.1. Ethics statement

This study was approved by the Habib University Ethics Review Committee, Karachi, Pakistan (HU- ERC-2025-AM1; March 2025). All participants were adults (18–30 years) who provided informed consent electronically (survey) or verbally (interviews). Participation was voluntary and judged minimal risk; sensitive questions were optional, withdrawal possible at any time, and resource sheets with local support services were provided. No direct identifiers or IP addresses were collected; interviews were recorded, securely stored, anonymized, and reported under pseudonyms.

De-identified data are retained on secure institutional storage and shared publicly via OSF. No direct compensation was provided beyond an optional prize draw (detailed below). The study was designed and conducted within a youth participatory action research (YPAR) framework.

### 2.2. Methodological rationale

Given the fundamentally exploratory nature of this study, we employ a convergent mixed methods design [21] to investigate how digitally connected urban youth in Pakistan repurpose GAI tools for health-related queries. This approach integrates a quantitative survey to establish patterns and predictors of GAI use with qualitative interviews to explore the motivations, perceptions, and experiences. We draw particular inspiration from socio-ecological youth health research from Mansfield and colleagues [19]. A pre-analysis plan for the quantitative analysis was registered at AsPredicted, #236264 (Wharton Credibility Lab). We used the GRAMMS Checklist [22] to ensure transparency, rigor and replicability (see S2 Fig).

### 2.3. Quantitative survey

The cross-sectional survey was developed through a multi-stage, iterative process informed by formative research, cognitive testing, pilot feedback, and a review of gray literature on digital health-seeking behaviors in LMICs. Care was taken to ensure that the survey was culturally appropriate, youth-centered, and mobile-optimized.

To minimize drop-offs and respondent fatigue, the final survey was limited to a median completion time of three minutes (see S1 Text for items). Key variables included demographics (age, gender, education board, sexual orientation), social support, healthcare avoidance, existing physical or mental health conditions, previous use of non-GAI health tools, trust in AI, confidence using AI, and awareness of AI risks. These were entered as predictors in multivariable logistic regression models assessing likelihood of GAI use for health. Bivariate associations (e.g., between gender and specific health-related queries) were tested using chi-square tests; model fit was assessed using McFadden's $R^2$, and multicollinearity was checked via generalized variance inflation factors (GVIF).

We used Google Forms to design the survey, given young people's familiarity with the UI/UX and in line with best practices in online survey research [23]. Data collection took place in June and July 2025, and the survey was disseminated through purposive and snowball sampling using digital platforms (e.g., WhatsApp, Instagram, X/Twitter, LinkedIn), targeting urban youth aged 18–30 across Pakistan. We also partnered with social media influencers (as suggested by [24]), local and international youth organizations, and universities (LUMS, IBA, HU, AKU, etc.) to post the survey on their platforms and disseminate via mailing lists (see Fig 2 for detailed strategy). A lucky- draw gift voucher worth PKR 5000 (≈18 USD) was advertised alongside the survey to nudge participation. The analytic sample (n = 1240; with events = 856) exceeded common rules-of-thumb (≥10–20 events per parameter) for multivariable logistic regression. Data were analyzed in R (version 4.3.2); all tests two-sided (α = .05). Missingness was handled via listwise deletion (≤3% per variable).

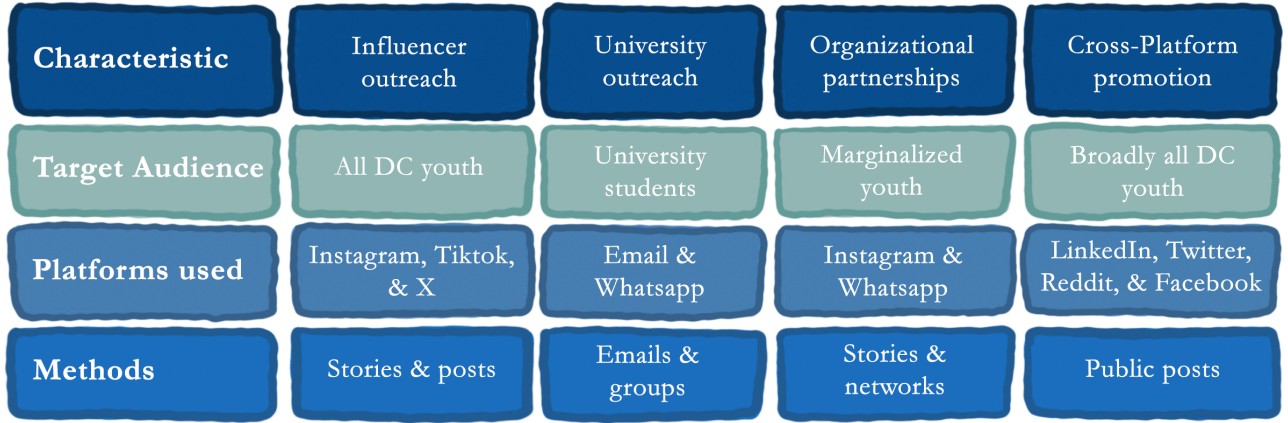

| Characteristic | Influencer outreach | University outreach | Organizational partnerships | Cross-Platform promotion |
|---|---|---|---|---|
| Target Audience | All DC youth | University students | Marginalized youth | Broadly all DC youth |
| Platforms used | Instagram, Tiktok, & X | Email & Whatsapp | Instagram & Whatsapp | LinkedIn, Twitter, Reddit, & Facebook |
| Methods | Stories & posts | Emails & groups | Stories & networks | Public posts |

**Fig 2. Quantitative survey dissemination strategy.**

### 2.4. Qualitative interviews

To complement the quantitative insights and gain richer data about youth GAI experiences, we conducted parallel semi-structured in-depth interviews with a diverse subset of participants (n = 20). The interview guide covered user motivations, trust in AI, the perceived role of AI in help-seeking journeys, and comparisons with peer, familial, or professional support systems. This interview guide was also informed by the same steps that helped create the quantitative survey.

Interviewees were selected via social media, i.e., individuals who responded to story and post requests and defined themselves as routine users of GAI for health. Interviews were conducted in June and July 2025, virtually, to ensure geographic accessibility and participant comfort. Recordings were transcribed verbatim via Turboscribe and analyzed using reflexive thematic analysis, guided by Braun and Clarke's six-phase approach [25]. The analytic process was iterative and inductive, with codes developed collaboratively by the research team and continuously refined to capture patterns across participant narratives. Emphasis was placed on preserving participant voice, especially in relation to agency, stigma, and lived experience concerning health and GAI.

### 2.5. Convergent design

Our convergent mixed-methods design was pre-specified and guided by GRAMMS, with concurrent timing and equal weighting of quantitative and qualitative components. Integration was embedded a priori across stages, e.g., formative qualitative work and a targeted literature review informed construct selection and item wording for both the survey and interview guide, while cognitive testing and a pilot survey fed back to refine qualitative sampling criteria, interview guides and prompts. For example, when early survey analyses showing higher GAI use among women and youth with mental or physical conditions led, we adapted the interview guide to probe more directly around gendered stigma, emotional burden, and how these young people decide when to use GAI, seek clinicians, (or both). Furthermore, throughout data collection, we used joint displays and narrative weaving during weekly team presentations to assess convergence, divergence, and other signals, which ultimately informed the development of a conceptual model, presented in the discussion section.

### 2.6. Reflexivity & Youth-Led research approach

This study was designed and implemented entirely by young Pakistanis aged 20–30 based in Habib University, in line with a youth-led participatory action research (YPAR) framework [26]. Such a research approach stands in contrast to traditional YPAR models that involve adult facilitation, which are most common in Pakistan.

This project was horizontally structured, with no older adult oversight besides receiving funding and IRB approval. Accordingly, youth researchers led all stages of the process, including (1) identifying research questions, (2) designing tools, (3) collecting data, (4) analyzing data, and (5) deciding how to best disseminate findings, as part of our commitment to epistemic justice and power-sharing ([26], pp. 402–406; see Fig 3).

To support research integrity and reflexivity, the team maintained a shared log documenting decision- making, dilemmas, and shifts in perspective. This reflexive infrastructure was a joint effort and was particularly important given the nature of the topic: youth health in Pakistan. Drawing on calls for stronger youth-centred accountability mechanisms (p. 414), we integrated regular peer check-ins and iterative revisions to our instruments and digital field strategy.

By centering youth as knowledge producers rather than subjects or informants, this study aligns itself with evidence that youth-led inquiry can strengthen research relevance, foster sociopolitical development, and enhance the translational impact of findings [Kim, 2016; 27,28,29]. In contexts where extractive research models dominate, especially in LMICs like Pakistan, this fully youth-led design offers an alternative model for ethical, locally grounded, and action-oriented public health scholarship (see: YPAR, Fig 3).

## 3. Results

Guided by our socio-ecological framework, we present the results in three linked stages. First, in Section 3.1, we report quantitative findings structured across ecological layers. Second, in Section 3.2, we present qualitative themes that map onto these same levels, but relatively iteratively (e.g., how participants describe individual agency, relational dynamics, and structural constraints in their health-seeking via GAI). Finally, in Section 3.3, we bring both these strands together in a convergent mixed-methods joint display, which visually integrates quantitative predictors and qualitative explanations across the socio-ecological model.

### 3.1. Quantitative results

1,305 individuals completed the survey. Out of these, 1,266 met the age criterion (18–30). Of these, 1,240 answered the primary outcome (analysis set). Table 1 shows the demographic breakdown; mostly using the age-eligible set (n = 1,266);

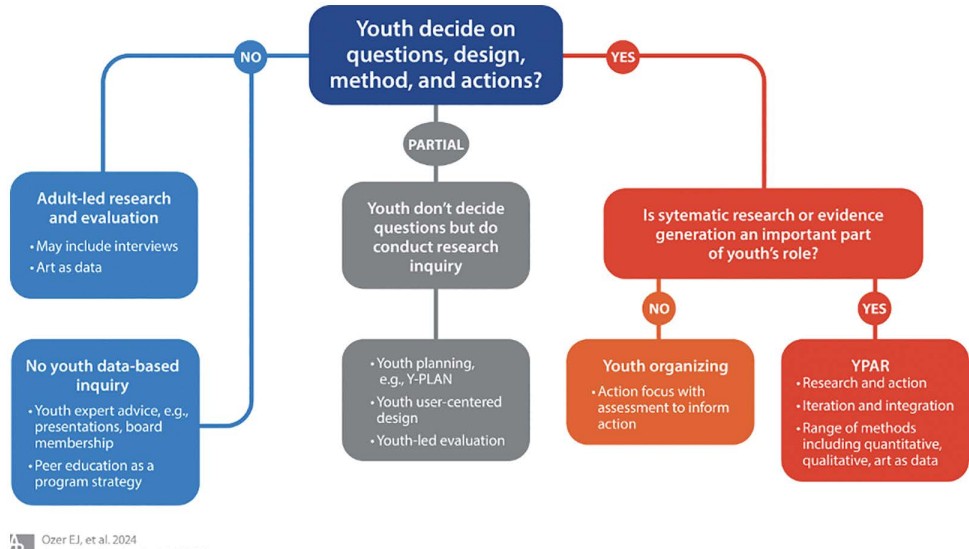

**Fig 3. YPAR Approach Framework [26].**

PLOS Digital Health

**Table 1. Sample Demographic Characteristics.**

| Variable | Category | n | pct |
|---|---|---|---|
| Age | Median: 22 [Q1: 20, Q3: 25] | | |
| Gender | Woman | 739 | 58.4 |
| | Man | 519 | 41.0 |
| | Other | 8 | 0.6 |
| Sexual Orientation | Heterosexual | 1,117 | 91.2 |
| | LGBTQ+ | 98 | 8.0 |
| | Other | 10 | 0.8 |
| Education Board | Local | 722 | 57.0 |
| | International | 532 | 42.0 |
| | Other | 12 | 0.9 |
| Close Friends | None | 87 | 6.9 |
| | 1-2 | 439 | 34.7 |
| | 3-4 | 490 | 38.7 |
| | 5+ | 249 | 19.7 |
| Comfort Discussing Health with Family | Yes | 442 | 35.0 |
| | Sometimes | 470 | 37.2 |
| | No | 350 | 27.7 |
| Existing Conditions | None | 661 | 52.4 |
| | Mental health | 194 | 15.4 |
| | Physical health | 189 | 15.0 |
| | Both | 217 | 17.2 |
| Non-GAI Health Tools | None | 920 | 72.7 |
| | Telemedicine | 126 | 10.0 |
| | WhatsApp/SMS | 89 | 7.0 |
| | Chatbots | 41 | 3.2 |
| | Other combinations | 90 | 7.1 |
| Used GAI for health | Yes | 856 | 69.0 |
| | No | 384 | 31.0 |
| General generative AI use (any purpose) | Daily | 687 | 54.3 |
| | A few times a week | 385 | 30.4 |
| | A few times a month | 73 | 5.8 |
| | Rarely | 95 | 7.5 |
| | Never | 26 | 2.1 |

Note: Age presented as median [Q1, Q3]; all others as n (%). Percentages use non-missing denominators per variable. 'Used GAI for health' (Yes/No) uses the Q1 analysis set (n = 1,240); 'General generative AI use (any purpose)' uses the age-eligible set (n = 1,266).

only the "Used GAI for health" row uses n = 1,240. Missing data were minimal, with the highest around ~3% for sexual orientation.

Overall, 69.0% (856/1,240) reported using GAI tools for health-related purposes, with ChatGPT being the most commonly used platform (96.1%, 820/853). The most frequent types of health-related queries were about physical symptoms (80.9%, 692/855), fitness or nutrition (59.9%, 512/855), and mental health (53.2%, 455/855). Key motivations included 24/7 availability (59.3%, 508/856), curiosity (52.8%, 452/856), and affordability (46.3%, 396/856). More than half of GAI users (57.6%, 491/853) reported feeling more comfortable asking sensitive health questions to AI than to a professional or doctor. Awareness of potential risks was also high: 72.0% (886/1,230) of all respondents acknowledged possible dangers

associated with AI use in health. Among them, 84.5% (747/884) were concerned about receiving inaccurate or unsafe advice, 55.0% (486/884) about developing over-dependence, and 48.3% (427/884) about privacy issues. Perceived usefulness was also high: 56.5% (481/851) rated AI-generated responses as "somewhat helpful," 39.2% (334/851) as "very helpful," and 4.2% (36/851) as "unhelpful."

The following bar chart (Fig 4) shows gender-disaggregated GAI use across health topics. Statistically significant gender differences were observed for mental health (p < .001), emotional support (p < .001), fitness (p = .049), and coping with a diagnosis (p = .026).

In our multivariable regression (Table 2), GAI use for health was significantly more likely among women (aOR = 1.57, 95% CI [1.17, 2.11], p = 0.003) and participants reporting any mental/physical health condition (aOR = 1.82, 95% CI [1.34, 2.48], p < .001). Higher trust in AI platforms strongly predicted use (aOR = 4.21, 95% CI [2.98, 6.01], p < .001). High confidence in using AI (aOR = 1.81, 95% CI [1.11, 3.07], p = 0.022), awareness of AI risks (aOR = 1.67, 95% CI [1.20, 2.31], p = 0.002), prior telemedicine use (aOR = 1.58, 95% CI [1.01, 2.54], p = 0.049), and use of other non-generative health tools (aOR = 4.48, 95% CI [2.59, 8.23], p < .001) were also associated with higher use.

By contrast, sexual orientation, type of education board (a proxy for social and economic class), perceived social support, and most categories of healthcare avoidance were not significantly associated with use. Participants who reported never avoiding healthcare had somewhat lower odds compared with those who often avoided care (aOR = 0.62, 95% CI [0.35, 1.10]), though this did not reach statistical significance (p = 0.102).

The model explained 14% of the variance (McFadden $R^2$ = 0.14), indicating moderate explanatory power. GVIF values suggested no evidence of multicollinearity, with all GVIF^(1/(2*Df)) ≤ 1.08. For further details, see S2 Table, S2 Text, S3 Fig, S3 Table and S4 Table.

### 3.2. Qualitative results

We analyzed 20 in-depth interviews with urban young adults aged 18–30 from across Pakistan. Participant pseudonyms are used to maintain anonymity. Consistent with reflexive thematic analysis, we judged the dataset to have adequate information power [30] for our aims; later interviews added nuance rather than new themes Below, we present the overarching themes explaining why young people turn to GAI tools for health. We draw on the COREQ checklist for reporting (S1 Table). Our reflexive memo and peer debriefings routinely evaluated the 1) methodological limitations, 2) coherence across data, 3) adequacy of data and 4) relevance to the review question.

**3.2.1. Theme 1: Access, affordability, and convenience.** Participants consistently emphasized how barriers such as increasing costs, long waiting times, and limited availability of quality services pushed them to turn to GAI tools as a

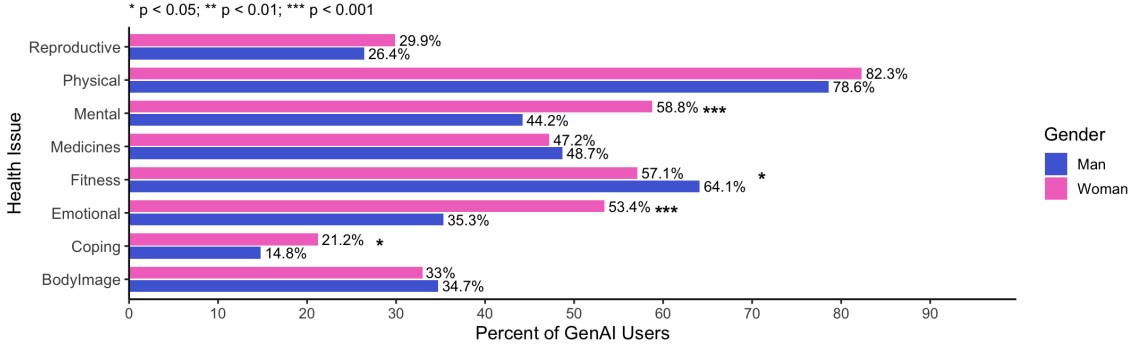

**Fig 4. Health issues disaggregated by Gender.** Note:. p < 0.05; ** p < 0.01; *** p < 0.001 (unadjusted chi-square/Fisher p-values).

**Table 2. Adjusted Odds Ratios for Regression Results.**

| Predictor | aOR_CI | p_fmt |
|---|---|---|
| Woman (vs. Man) | 1.57 [1.17, 2.11] | 0.003 ** |
| LGBTQ+ (vs. Heterosexual) | 0.59 [0.34, 1.05] | 0.068 |
| Other (vs. Heterosexual) | 2.61 [0.35, 53.47] | 0.410 |
| International board (vs. Local) | 1.12 [0.83, 1.51] | 0.455 |
| Other board (vs. Local) | 0.40 [0.07, 1.80] | 0.245 |
| Social support: High (vs. Low) | 0.91 [0.65, 1.29] | 0.601 |
| Any condition (Yes vs. No) | 1.82 [1.34, 2.48] | <.001 *** |
| Healthcare Delay: Never (vs. Often) | 0.62 [0.35, 1.10] | 0.102 |
| Healthcare Delay: Rarely (vs. Often) | 0.91 [0.62, 1.35] | 0.647 |
| Healthcare Delay: Sometimes (vs. Often) | 0.92 [0.63, 1.34] | 0.678 |
| Trust in AI (per level) | 4.21 [2.98, 6.01] | <.001 *** |
| Confidence: High (vs. Low) | 1.81 [1.11, 3.07] | 0.022 * |
| Aware (vs. Not aware) | 1.67 [1.20, 2.31] | 0.002 ** |
| Telemedicine user (vs. None) | 1.58 [1.01, 2.54] | 0.049 * |
| Other tool user (vs. None) | 4.48 [2.59, 8.23] | <.001 *** |

Note: Significance: *** $p < .001$; ** $p < .01$; * $p < .05$. McFadden $R^2 = 0.14$. All GVIF^(1/(2*Df)) ≤ 1.08.

resource. GAI (specifically, ChatGPT) was perceived as always accessible and "there for you" when health services were not. This was our most potent and consistent finding, across all interviews.

As Ayesha explained, "[...] therapy is definitely something that's available but at the same time it's not something that's available 24/7 [...] AI has been very helpful in that sense. At 3 AM, no therapist is available, but I have ChatGPT, for example, to kind of keep track of all my emotions." This constant availability was important for participants who: 1) struggled to secure appointments, 2) could not afford consultations, and 3) had previously had adverse health experiences with professionals.

As one key informant noted, "Usually in Pakistan [...] we go to hospitals when we're about to die [hyperbole]. Because our health system is not that good [...] People are afraid to go to the doctor, who will bear the expenses?" (Junaid). This quote illustrates what many participants highlighted, i.e., that healthcare- seeking involved significant out-of-pocket expenditure, particularly in private facilities, leading them to either delay or avoid consultations altogether.

Zashe echoed the challenges of long waiting times in addition to cost: "Waiting times at hospitals are so long. ChatGPT provides immediate answers which is reassuring." This immediacy allowed participants to triage their needs, self-manage minor issues, or decide whether in-person care was necessary. Mujahid summarized this sentiment: "Healthcare is costly and sickness is common [...] AI gives you some answers when going to the doctor isn't possible."

**3.2.2. Theme 2: Emotional safety and support.** A second theme centered on the emotional safety and informational value that participants derived from GAI interactions, overwhelmingly with ChatGPT.

GAI tools were described as judgment-free, non-stigmatizing ("doctors here [in Pakistan] are very fatphobic"), and capable of providing sensitive and holistic explanations about health concerns that professionals in Pakistan were perceived to often lack the time or training (or both) to deliver. Several participants described AI as a space where they could express sensitive issues such as sexual health, mental health, neurodivergence, or weight concerns, without fear. One female participant noted, "AI can help with topics [...] like sexual health or weight issues where I have been shamed by doctors" (Samina). This sentiment was echoed by Aliza: "The stakes are obviously much lower. It's just AI. It's not like it's going to tell anyone" and others: "there's always this period, taking sessions, figuring out how secretly homophobic or slut-shamey my therapist is".

Several participants also valued the absence of anxiety and nervousness that otherwise exists with human interaction, even for non sensitive issues, e.g., ability to ask multiple questions without embarrassment or time constraints. As Mohib puts it: "With ChatGPT, there's no fear of judgment like there is when sharing sensitive information with a therapist [...] I can ask the same question five different ways without feeling awkward or weird.". Other respondents noted similarly: "Even if the responses are biased, sometimes it's enough to give you closure [...] it gives some peace."; "Sometimes I don't want a solution. I ask, are my feelings valid?."

Beyond emotional safety, GAI was also perceived as a rich source of information that participants struggled to access from healthcare professionals. "Doctors don't really tell you what's up [...] they're busy and won't engage with you that much. ChatGPT helps me understand my situation better." Similarly, Saad described using AI to clarify medication information: "In Pakistan, doctors tend to be a bit cruel. They're not going to tell you everything [...] But ChatGPT told me exactly what the medicines were for and how to take them."

**3.2.3. Theme 3: Empowerment and agency in navigating the health process.** Finally, participants described using GAI tools as a way to gain agency and greater control over their health-seeking journeys. They reported feeling more empowered to interpret medical content (e.g., ulcer reports), organize their concerns, and communicate more effectively with professionals.

Asad described how GAI helped him navigate complex test results: "I used GPT to interpret a tricky [name of scan and details redacted] report for my mother." Similarly, Marij explained how GAI enabled him to frame health concerns for appointments: "AI even gives me the words to articulate my health issue to the doctor in a structured way, otherwise they just dismiss you for rambling." Similarly, other key informants noted the following: "I trained my GPT to sympathize first or offer solutions later, depending on what I need"; "I use the same chat so it 'remembers' meds and side effects, it's like a running log for my medications."

Furthermore, this sense of empowerment extended beyond technical knowledge. Several participants described using GAI to track their symptoms, set health goals, and self-manage conditions. Ghazi explained: "I use ChatGPT to track medications, supplements, and side effects [...] I use it for calorie deficits and screen addiction; it gives me a roadmap."

GAI tools also offered a more holistic perspective on health than participants perceived from doctors. As Marij observed: "AI takes a more holistic view… doctors in Pakistan mostly just focus on symptoms." Participants particularly appreciated AI's ability to link information across previous conversations (e.g., the memory feature that remembers information) and highlight overlooked patterns, which reinforced their sense of control and preparedness.

Overall, across themes, participants framed GAI tools as accessible, non-judgmental, and empowering complements to an overburdened healthcare system. For many, these platforms represented the only viable way to seek health-related information and emotional support without significant cost, stigma, or delay. Furthermore, while participants emphasized that GAI should not replace healthcare professionals, it was often the first point of contact for a variety of health related concerns. This was especially true for sensitive and stigmatized topics.

## 3.3. Data integration

As recommended by Creswell & Plano Clark [21], we present a joint display that brings together key quantitative predictors and qualitative themes. While the survey data identify significant predictors of GAI use, the interviews help describe and explain why these associations matter in practice. The table below (Table 3) highlights points of convergence and divergence among key predictors.

Anchoring the joint display in the socio-ecological model provides a full narrative of GAI use for health by young people, i.e., at the individual level, higher odds for women, any condition, and strong trust/confidence driving uptake. Our formative research suggested this too. At the relational level, the social support variable is non-significant, yet interviews show GAI as a workaround when family/peer spaces feel unsafe or shaming suggesting standard support measures

**Table 3. Convergent design visual display.**

| Predictor | aOR [95% CI] | p-value | Illustrative Quotes | Integration & Meta-Inference |
|---|---|---|---|---|
| **Individual-Level Factors** | | | | |
| Woman (vs. Man) | 1.57 [1.17, 2.11] | .003 ** | "So I feel like women are definitely more open to it, just like women are more open to therapy. Again, when it comes to talking about feelings, I feel like feelings or things like that, I feel like that's just a general theme that women are usually more open to it. But I've also seen a lot of men eventually open up to the idea when they see that it actually helps." | Convergence. Higher reported use by young women converges with qualitative data on women's open-ness to seek help, espe-cially for issues relating to mental health, emotions, and gendered concerns (also see Fig 4). |
| International board (vs. Local) | 1.12 [0.83, 1.51] | .455 | | Convergence: schooling-type, and economic status, within the urban population was not considered important. Instead, the 'ability' to prompt correctly, was considered critical to meaningful and sustained GAI use. However the null board effect should not be read as evidence that class or inequality are unimportant; rather, it may indicate that educational board type was possibly too blunt a proxy to capture the more specific forms of structural advantage that matter here, such as English fluency, device quality, algorithmic familiarity, connectivity, etc. |
| LGBTQ+ (vs. Heterosexual) | 0.59 [0.34, 1.05] | 0.068 | "So I used to go to AI for questions related to symptoms of different STDs, how do they spread, why do they spread, is it normal, is it okay, does the | Divergence: Quantitative data shows status as sexual orientation as a non-significant predictor. However, qualitative interviews routinely |
| | | | society accept, so there were these small questions, basically" | emphasised its utility for marginalised groups, including the LGBTQ+ community. |
| Any condition (Yes vs. No) | 1.82 [1.34, 2.48] | <.001 *** | "I use ChatGPT to track medications, supplements, and side effects, it gives me a roadmap." "But in terms of my ADHD and how it helps me manage. And, you know, reduce my stress and burnout." | Convergence. Past conditions (mental or physical, or both) converged with qualitative reports of symptom management and constant monitoring. |
| **Relational Factors** | | | | |
| Trust in AI (per level) | 4.21 [2.98, 6.01] | <.001 *** | "ChatGPT told me exactly what the medicines were for and how to take them." "So I can confidently say that the accuracy of these plat-forms and the information they offer you, it's quite accurate." | Convergence. Trust amplifies use; qualitative accounts show perceived clarity and usefulness driving usage. Users reported triangulating ChatGPT responses with doctors' diagnoses and prescriptions/advice. |
| Confidence: High (vs. Low) | 1.81 [1.11, 3.07] | .022 * | "I've created a code word and it saves everything that I tell it under that code word related to that topic. So anytime that I needed to pull that information up and relate to that and go back on it, it kind of pulls that up." | Convergence. Prompting skill/comfort lowers friction and sus-tains engagement; supports targeted "how to use AI for health" micro- modules. |
| Aware of AI risks (Yes vs. No) | 1.67 [1.20, 2.31] | .002 ** | "I feel like it is a bit biased. It just tells you things that you want to hear. So in that category, I wouldn't say it's that human." | Convergence/Paradox. Awareness coexists with use; adopt harm-reduction practices (verification, using AI as a "first step"), implying scope for formal risk-literacy. |
| High Social Support (vs. Low) | 0.91 [0.65, 1.29] | 0.601 | I belong to a very conservative family, where a lot of discus-sions regarding sexual health, a lot of discussions regarding basic health, are not really openly discussed…with regards to me being a man, my body changing, or me going through problems, I would have to figure it out on my own | Divergence: Quantitative data suggests perceived social support was not significant (friends and family). Qualitative data, too, showed variety, i.e., while some individuals turned to GAI due to lack of support, some with very good relationships still found it a useful tool. |

*(Continued)*

Table 3. (Continued)

| Predictor | aOR [95% CI] | p-value | Illustrative Quotes | Integration & Meta-Inference |
|---|---|---|---|---|
| Community-Level Factors | | | | |
| Telemedicine user (vs. None); Other tool user (vs. None) | 1.58 [1.01, 2.54] 4.48 [2.59, 8.23] | .049* <.001 *** | [talking about sexual health as a queer person: "Before AI I used Reddit, but with AI I didn't have to post publicly or leave a digital footprint. I could just chat and delete." "I've typically tried using mental health apps [smartphone applications] that were solely based on AI." | Divergence/Mixed Signals: Many participants in the qualitative sample had never used any tool for health, suggesting accessibility and interface lead to exploration of health related issues. |
| Healthcare Avoidance/ Delay | 0.62 [0.35, 1.10] | .102 | "We all know that we don't have that many therapists in Pakistan anyway. But [...] it ends up being for some time next week, sometime a few days later when the moment has passed. You're having anxiety attack right now" | Divergence: Quantitative Data suggests no meaningful relationships, but participants both implicitly and explicitly referenced how GAI use might fill in structural healthcare inefficiencies in Pakistan. |

**Note:** Convergence marked in green; Divergence marked in red

miss quality/safety of ties. Finally, at the community/system level: telemedicine, other tool use, and narratives of cost, wait times, and access indicate a digital-first ecology amid service constraints; taken together with qualitative accounts of prompting skill, English fluency, and prior platform familiarity, these findings suggest that reported uptake is concentrated among early adopters who might be simultaneously: structurally constrained by health-system failures and, still, relatively privileged within the digital landscape.

## 4. Discussion

This study provides early LMIC estimates of youth engagement with GAI for health, showing that nearly 70% of urban Pakistani youth report use. Uptake was patterned by gender, pre-existing health conditions, and trust in AI. Our findings stand in sharp contrast to recent figures from high-income countries (HICs), where uptake remains far lower (e.g., 11% in the United States, [2]). While such cross-context comparisons are only illustrative, given differences in sampling and the infancy of this field, they still demonstrate the need and urgency to understand the distinctive structural and cultural drivers shaping adoption in LMICs. We hope this exploratory, mixed-methods study can help researchers generate and test new hypotheses related to youth appropriation of GAI tools. Our results are synthesized and mapped visually in Fig 5. This conceptual model locates both spatially (ecological layers) and temporally (ordered events) the use of GAI for health support youth in the Pakistani context. In the following discussion we: 1) interpret findings through four interrelated themes, before turning to 2) methodological reflections, strengths and limitations, and implications for policy and practice.

Importantly, these findings should not be read as representative of Pakistani youth as a whole. Rather, they likely capture a leading-edge subgroup of urban, digitally connected, English-literate young people who are relatively more exposed to GAI platforms and better positioned to experiment with them. In that sense, this cohort may be understood less as a population estimate of youth health-seeking in Pakistan and more as an early adoption group whose practices may foreshadow wider diffusion, while also reflecting existing structural privilege. This may explain both the high reported prevalence and the specific forms of use observed in our data.

### 4.1. Health gaps, stigma, and compensatory use

Youth living with pre-existing health conditions were twice as likely to use ChatGPT for health, showing how GAI may serve as a compensatory resource where formal services are absent and/or insufficient. Particularly notable were sociocultural barriers, i.e., stigma surrounding sexual and mental health shaped both the content and nature of engagement, echoing prior work on health- seeking in Karachi [10]. Women were especially likely to use GAI, consistent with research on higher baseline help-seeking among women [31,32], the influence of gender norms [33,14,18], and the scarcity of confidential, nonjudgmental services in Pakistan [9]. Taken together, our findings highlight how GAI help-seeking may be, in many cases, a necessary workaround for deeply embedded structural and cultural barriers to care, especially in the LMIC context.

### 4.2. Emotional safety and affective support

Another key theme was the emotional utility of GAI tools. Participants described ChatGPT as a supportive, anonymous presence that helped them navigate late-night anxiety, relational conflict, and identity dilemmas. This mirrors emerging global research on GAI's ambient mental health functions [6,34], but is particularly notable in Pakistan's context of stigma, silence, and familial surveillance [13,8]. Some participants likened the tool to a "friend" or "therapist," validating their concerns in a nonjudgmental tone. Others emphasized its value as a first step, something to consult before making decisions, particularly when human help felt inaccessible (Fig 5). These findings resonate with literature documenting how users treat AI- enabled tools as relational supports during periods of distress [16,35]. Yet participants flagged risks: feelings of over-reliance, the lack of

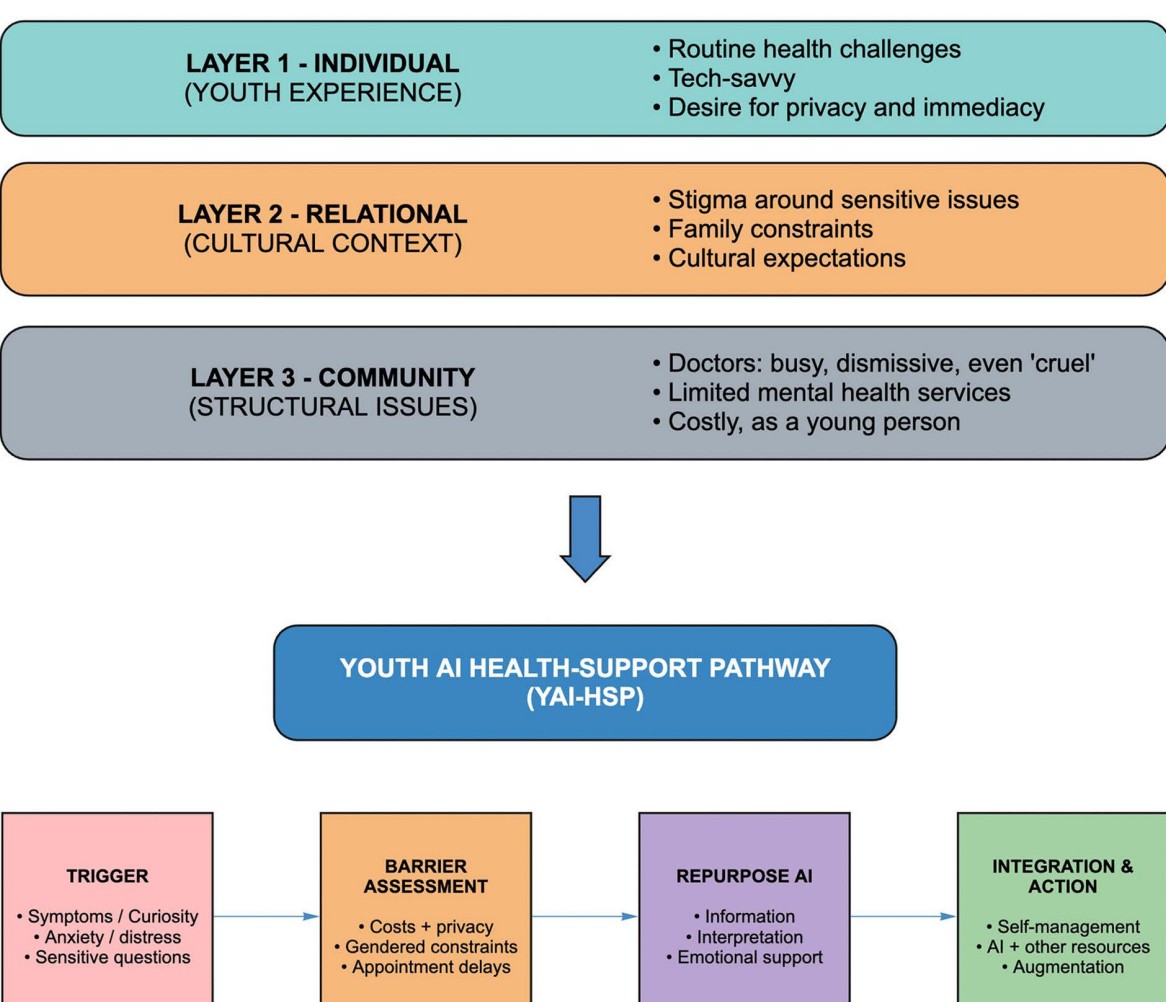

**Fig 5. YAI-HSP Conceptual Model.**

accountability, and concern about engaging too deeply with a non-human interlocutor [36]. Such tensions present an important ethical and psychological consideration for public health actors in LMICs, where relational care is both scarce and overburdened [37].

### 4.3. Inequalities of access and AI literacy

While many participants engaged actively with ChatGPT, iteratively refining prompts to interpret lab results, understand diagnoses, prepare for clinical encounters, etc; such empowerment is unlikely to be evenly shared. Youth with stronger forms of cultural capital, such as English proficiency and prior digital confidence, were more likely to use and benefit from

GAI: Interviewees described developing "AI literacy": the ability to engineer prompts and critically evaluate outputs. Yet in Pakistan, where fewer than 2% of people pursue higher education and private, English-medium schooling remains a major class divide [38], such skills, or the ability to build them quickly, are unequally distributed. This aligns with the concept of cultural health capital [39] and suggests that GAI may widen, rather than narrow, existing inequities.

Out reflexive memos demonstrate this best: our research originally considered the possibilities, as narrated by our participants, for GAI to democratize health access and resources. However, over the course of data collection and our sustained reflection, we found that rather than flattening inequalities, GAI appeared most useful to those already equipped with a certain cultural capital (language, cultural knowledge, confidence, digital literacy, platform familiarity, etc.) to ask the 'right' questions; to articulate emotions/pain in a language that LLM models were trained to understand, and to cast doubt when necessary. It was this cultural health capital that could allow them to 'prompt engineer' most usefully, and engage in conversational querying.

Our attempt to capture socioeconomic status (SES) via educational board type (international vs. local) was informed by formative research but has limitations. While the education board reflects some classed dimensions of cultural capital, it may not comprehensively capture precarity linked to household insecurity or intergenerational wealth. Future youth-led work should experiment with and employ more holistic SES measures that better reflect the class divides in Pakistan.

### 4.4. Convenience, personalization, and cost

Participants repeatedly cited GAI's immediacy, no/low cost, and user-friendly tone as critical factors enabling use. ChatGPT was preferred over Google because it was more conversational, less overwhelming, and available on demand. These findings showcase GAI platforms' appeal as a responsive, low-barrier tool, especially in a context like Pakistan, where adolescent-friendly services are scarce and healthcare costs are prohibitively high [8,10]. Frustrations with local healthcare were in line with expectations and literature [40].

The ability to tailor prompts and receive personalised responses was described as empowering. However, participants also acknowledged the risk of "false confidence" i.e., being misled by GAI's fluent style and apparent authority. Both the survey and interviews emphasized the importance of verifying GAI responses with human experts. This pragmatic approach, of treating ChatGPT as "a step up from Google" rather than a doctor, echoes recent calls for hybrid care models where informal tools augment formal systems instead of replacing them (see Integration, Fig 5).

### 4.5. Ethical awareness, misinformation, and climate concerns

Despite high levels of engagement, participants demonstrated considerable awareness of GAI's limitations. Risks cited included misinformation, culturally inappropriate advice, affirmation bias, and limited relevance for specific medical conditions. These concerns resonate with recent research cautioning against over-reliance on GAI tools in health decision-making [41] and many of the risks pointed out were comparable to those of professionals [42].

A surprising and rather unprompted theme was climate anxiety. Although our survey did not include climate concerns, participants used the open-ended "other concerns" field to express concerns about GAI highlighting a form of digital conscientiousness rarely captured in public health literature. Given Pakistan's history, vulnerability to climate change, and the rise of climate-oriented youth movements, this finding merits further exploration in the LMIC context [5].

### 4.6. Methodological strengths & reflexive insights

We argue that this study's methodological design is itself a key contribution in the context of LMIC YPAR research; for instance, the survey instrument was developed through rigorous formative research guided by socio-ecological theory, including cognitive testing to ensure clarity and brevity (≤3 minutes). Design choices were shaped by the constraints of

balance: i.e., online youth data collection (ensuring minimal drop-off, higher N) and construct validity of survey-items, with all decisions deliberated up, tested, co-produced, within a resource-scarce setting.

Interestingly, the process also revealed structural barriers to youth participation, e.g., the inclusion of a sexual orientation question, essential to public health equity, triggered backlash from some respondents and deterred participation from more conservative male social media influencers. Also, open-text in the survey responses included hostile remarks such as, "this is a muslim country," illustrating the persistent stigma even within 'elite', urban youth spaces. We believe this may be among the first locally led surveys in Pakistan to include sexual orientation in a quantitative public health context, and we call for more ethical yet courageous efforts to ensure sexual and gender minority inclusion in future research [43,44,45].

### 4.7. Strengths & limitations

We claim multiple strengths in this study, which include a pre-specified convergent mixed-methods design with equal weighting, youth-led design, which includes instrument development with cognitive testing/piloting, a relatively large analytic sample (N = 1,240), transparent reporting (pre-analysis plan, model fit, multicollinearity checks), and integration via joint displays that link adjusted odds ratios to interview themes. Perhaps most importantly, this is amongst one of the first and few studies that look into this topic (youth GAI use) using this epistemic approach (YPAR).

We also acknowledge several natural limitations, which include non-probability online recruitment (purposive/snowball), introducing potential sampling bias and restricting generalizability to rural Pakistan; an English-literate, tech-savvy, digitally connected, largely urban sample that likely systematically overestimates both exposure to and adoption of GAI for health relative to the broader youth population in Pakistan. Recruitment through online platforms, universities, youth organizations, and social media networks may have preferentially reached participants who were already more comfortable with digital tools, more familiar with GAI platforms, and more likely to view them as acceptable sources of information. While targeting digitally connected youth was intentional, to uncover trends in early adoption and potential trickle-down, future research should expand our work and focus on more diverse youth populations.

Taken together, our results should be read as descriptive evidence among digitally connected urban youth in Pakistan, with a mixed-methods integration and reflexive YPAR approach to strengthen credibility and policy relevance/practice. We hope that future researchers can build on this evidence and also continue incorporating YPAR in the LMIC health context.

### 4.8. Policy and practice implications

Health systems must plan for the emerging reality that GAI may become a de facto first point of contact for many young people. To ensure that this use enhances rather than undermines wellbeing, our findings point to five interconnected policy and practice implications:

1. **Integrate into digital health plans:** Acknowledge youth GAI use in e/mHealth and track risks and opportunities with clear indicators.

2. **Treat AI literacy as health literacy:** Teach critical appraisal in schools and community programs (e.g., PMYP/Generation Unlimited).

3. **Train providers in digital empathy:** Equip frontline staff to respond nonjudgmentally to the needs driving GAI use and strengthen therapeutic alliances.

4. **Set equity-first guardrails:** Require cultural/linguistic inclusion and transparency; co-design with youth to reduce disparities.

5. **Back youth-led research & advocacy:** Recognize GAI as a structural determinant of health information-seeking and embed findings in national and WHO-aligned strategies.

## 5. Conclusion

Using a theory-driven framework and youth-led design, we identify who uses GAI for health, how, and why. We show that GAI, such as ChatGPT, are functioning as stopgaps, emotional scaffolds, and health literacy aids in the absence of responsive formal care. Youth in Pakistan are turning to them not because they are ideal, but because they are available, affordable, and anonymous, reflecting both the potential and the peril of unregulated GAI in under-resourced settings. Our work hopes to raise critical questions about this new form of therapeutic alliance, augmented by GAI tools, and we call on public health systems and actors to consider this emergent phenomenon more seriously, including by ethically integrating GAI into LMIC digital health strategies, monitoring its use within national e/mHealth agendas, and co-designing safeguards with youth who are already using these tools. Future research (in Pakistan, and elsewhere in the Global South) should build on this hypothesis-generating, exploratory work by drawing on larger and more diverse samples, including rural and non–English speaking youth, and by using designs that can more rigorously examine causal pathways and compare uptake patterns with HIC settings.

### Patient and public involvement

Youth were involved in all stages (question development, piloting, recruitment materials, interpretation, dissemination) consistent with a YPAR approach.

### Supporting information

**S1 Fig. Worldwide search results for "ChatGPT" by country via Google Trends.**
(PDF)

**S1 Text. Full quantitative survey instrument and analysis variables.**
(PDF)

**S2 Fig. Good Reporting of a Mixed-Methods Study checklist.**
(PDF)

**S1 Table. COREQ 32-item checklist.**
(PDF)

**S2 Text. Additional quantitative analyses and diagnostics.**
(PDF)

**S2 Table. Trust in AI sensitivity analyses.**
(PDF)

**S3 Table. Interaction models.**
(PDF)

**S4 Table. Missing data by variable.**
(PDF)

**S3 Fig. Calibration plot for the final regression model.**
(PDF)

## Acknowledgements

**Generative AI Use:** During the preparation of this work, the authors used GPT to improve word choice and debug coding errors. After using this tool and service, the authors reviewed and edited the content as needed and take full responsibility for the content of the publication.

## Author contributions

**Conceptualization:** Ahsan Mashhood, Aamna Ahmed, Inaya Khan, Maryam Hashim, Sara Baloch.

**Data curation:** Ahsan Mashhood, Aamna Ahmed, Inaya Khan, Maryam Hashim, Sara Baloch.

**Formal analysis:** Ahsan Mashhood, Aamna Ahmed, Inaya Khan, Maryam Hashim.

**Funding acquisition:** Ahsan Mashhood.

**Investigation:** Ahsan Mashhood, Aamna Ahmed, Inaya Khan, Maryam Hashim, Sara Baloch.

**Methodology:** Ahsan Mashhood, Aamna Ahmed, Inaya Khan, Maryam Hashim, Sara Baloch.

**Project administration:** Ahsan Mashhood, Aamna Ahmed, Inaya Khan, Maryam Hashim.

**Resources:** Ahsan Mashhood.

**Software:** Ahsan Mashhood.

**Supervision:** Ahsan Mashhood.

**Validation:** Ahsan Mashhood.

**Visualization:** Ahsan Mashhood.

**Writing – original draft:** Ahsan Mashhood.

**Writing – review & editing:** Ahsan Mashhood, Aamna Ahmed, Inaya Khan, Maryam Hashim, Sara Baloch.

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
