## [Decision Letter · Decision Letter 0]

17 Nov 2025

Response to Reviewers '. This file does not need to include responses to any formatting updates and technical items listed in the 'Journal Requirements' section below.* A marked-up copy of your manuscript that highlights changes made to the original version. You should upload this as a separate file labeled 'Revised Manuscript with Track Changes '.* An unmarked version of your revised paper without tracked changes. You should upload this as a separate file labeled 'Manuscript '. If you would like to make changes to your financial disclosure, competing interests statement, or data availability statement, please make these updates within the submission form at the time of resubmission. Guidelines for resubmitting your figure files are available below the reviewer comments at the end of this letter. We look forward to receiving your revised manuscript. Kind regards, Umbereen Sultana Nehal Guest EditorPLOS Digital Health Harry HochheiserSection EditorPLOS Digital Health Leo Anthony CeliEditor-in-ChiefPLOS Digital Healthorcid.org/0000-0001-6712-6626  **Journal Requirements:**

1. Please ensure that your Ethics Statement is available in its entirety at the beginning of your Methods section, under a subheading 'Ethics Statement'. It must include:

1) The name(s) of the Institutional Review Board(s) or Ethics Committee(s)

2) The approval number(s), or a statement that approval was granted by the named board(s)

3) (for human participants/donors) - A statement that formal consent was obtained (must state whether verbal/written) OR the reason consent was not obtained (e.g. anonymity).

2. We ask that a manuscript source file is provided at Revision. Please upload your manuscript file as a .doc, .docx, .rtf or .tex.

3. Please upload separate figure files in .tif or .eps format. Also, remove the figures from your manuscript file but keep the legends.

4. Please provide an Author Summary. This should appear in your manuscript between the Abstract (if applicable) and the Introduction, and should be 150–200 words long. The aim should be to make your findings accessible to a wide audience that includes both scientists and non-scientists. Sample summaries can be found on our website under Submission Guidelines:

https://journals.plos.org/digitalhealth/s/submission-guidelines#loc-parts-of-a-submission

5. Some material included in your submission may be copyrighted. According to PLOS’s copyright policy, authors who use figures or other material (e.g., graphics, clipart, maps) from another author or copyright holder must demonstrate or obtain permission to publish this material under the Creative Commons Attribution 4.0 International (CC BY 4.0) License used by PLOS journals. Please closely review the details of PLOS’s copyright requirements here: PLOS Licenses and Copyright. If you need to request permissions from a copyright holder, you may use PLOS's Copyright Content Permission form.

Potential Copyright Issues:

a. Figure S1.2: please (a) provide a direct link to the base layer of the map (i.e., the country or region border shape) and ensure this is also included in the figure legend; and (b) provide a link to the terms of use / license information for the base layer image or shapefile. We cannot publish proprietary or copyrighted maps (e.g. Google Maps, Mapquest) and the terms of use for your map base layer must be compatible with our CC-BY 4.0 license.

* PlaniGlobe - All maps are published under a Creative Commons license so please cite “PlaniGlobe, http://www.planiglobe.com, CC BY 2.0” in the image credit after the caption. (http://www.planiglobe.com/?lang=enl

b. Figure S2: Please confirm whether you drew the images / clip-art within the figure panels by hand. If you did not draw the images, please provide (a) a link to the source of the images or icons and their license / terms of use; or (b) written permission from the copyright holder to publish the images or icons under our CC-BY 4.0 license. Alternatively, you may replace the images with open source alternatives. See these open source resources you may use to replace images / clip-art

- https://openclipart.org/

**Additional Editor Comments (if provided):** Thank you for your patience with the length of time this manuscript with with us -- it has been challenging to get reviewers of late. We are pleased to offer to accept this manuscript with minor revisions only.

Two of the three reviewers suggested major revisions, but that was looking purely at the model by itself, whereas your community participatory research has both qualitative and quantitative elements. The co-design with youth as well as the numerous direct quotes -- this multi-model approach to using several types of data and analyses to get multiple views on the issue -- mitigate some of the reviewer concerns related to the model alone. The reviewers comments regarding need for certain rigor would be more weighty if the manuscript were presenting the model as a stand alone rather than as supportive to the qualitative research aspects.

The reviewer comments on the sampling methods, which reference best practices in research, can be addressed by stronger language, in addition to what you already have regarding generalizability. For a population that is online and using tech tools or AI for health, the sampling and recruitment choice seems reasonable.

Also, as your research is in a less explored area, it falls into hypothesis generating and exploratory. Please incorporate this into your revision to ensure that the introduction and discussion highlight the multi-modal analysis, the hypothesis generation aspects, the exploratory nature, and how this will be the initial work to inform future work.

**Reviewers' Comments:** Reviewer's Responses to Questions

**Comments to the Author**

1. Does this manuscript meet PLOS Digital Health’s publication criteria ? Is the manuscript technically sound, and do the data support the conclusions? The manuscript must describe methodologically and ethically rigorous research with conclusions that are appropriately drawn based on the data presented.

Reviewer #1: Yes

Reviewer #2: Yes

Reviewer #3: Yes

2. Has the statistical analysis been performed appropriately and rigorously?

Reviewer #1: Yes

Reviewer #2: Yes

Reviewer #3: Yes

3. Have the authors made all data underlying the findings in their manuscript fully available (please refer to the Data Availability Statement at the start of the manuscript PDF file)?

Reviewer #1: Yes

Reviewer #2: Yes

Reviewer #3: No

4. Is the manuscript presented in an intelligible fashion and written in standard English?

Reviewer #1: Yes

Reviewer #2: Yes

Reviewer #3: Yes

Reviewer #1: Major comments

Framing and novelty

The manuscript convincingly positions itself within early LMIC evidence on GAI in health, but claims of being “first” should be softened to “among the first.” Limit claims to digitally connected urban youth to reflect the actual sampling frame.

Sampling and bias

Recruitment through social-media and campus networks may over-represent tech-savvy respondents. The manuscript should describe duplicate-entry safeguards (timestamps, de-duplication by contact info, CAPTCHA, etc.) and provide city/region distribution and recruitment-to-completion rates. If feasible, include a sensitivity check or demographic weighting showing whether the 69 % prevalence remains stable.

Outcome definition

Clarify exactly how “use of generative AI for health” was measured: include the wording of survey items, response options, and the recall period (e.g., “ever,” “past 12 months”). The precise definition is crucial for interpreting the reported 69 % figure and topic-specific patterns.

Model specification

The multivariable logistic regression is appropriate but needs fuller reporting.

• Describe the scale and linearity assumption for ordinal predictors such as “trust in AI.” If necessary, test non-linearity or use categorical coding.

• Add AUC, calibration plot, and Brier score alongside McFadden’s R².

• Consider key theory-driven interactions (e.g., gender × condition, trust × confidence).

• Report the exact number of parameters and events-per-parameter ratio.

• The ICC confidence interval extending beyond 1.0 must be corrected.

Missing data

Listwise deletion for ≤3 % missingness is acceptable, but summarize missingness by variable and, if possible, show that results are robust to simple imputation.

Qualitative rigor

The interviews focus on self-identified routine users, which may bias themes toward pro-use narratives. Explicitly acknowledge this and note any dissenting or negative cases. Include a brief participant-characteristics table (age, gender, city, prior digital-health experience).

Integration clarity

The joint-display integration is effective. Strengthen it with one concrete example of how a quantitative finding guided qualitative questioning or vice versa, to demonstrate procedural—not just interpretive—mixing.

Language and causal tone

Replace causal verbs with associative phrasing (“linked to,” “associated with”). In the implications section, offer concrete harm-reduction practices for youth (e.g., verifying AI health advice, seeking clinician confirmation) and touch on governance needs for local platforms.

Ethics and openness

Ethics approval and OSF sharing are properly noted. Place the OSF link prominently in the Data-Availability statement and include a short description of what is hosted there (dataset, survey, codebook).

Minor comments

Specify exact anchors for the “trust” and “confidence” scales and note whether they were adapted from validated tools.

In regression tables, clearly mark reference groups and show non-missing n per variable.

Provide a simple participant-flow diagram (views → starts → completes → analyzed).

Label multiple χ² comparisons as exploratory or apply false-discovery-rate control.

referring to these studies will enhance your argument

https://doi.org/10.1186/s12909-024-05406-1

https://doi.org/10.1007/s00261-025-04810-5

Minor proofreading: eliminate repeated phrases and ensure consistent tense

Reviewer #2: The manuscript in title “Use of Generative AI for Health Among Urban Youth in Pakistan: A Mixed-Methods Study” explores a highly relevant and emerging topic—the use of generative AI for health among urban youth in Pakistan. The study is well-designed, with a solid mixed-methods approach and clear theoretical grounding in the socio-ecological framework. It offers valuable early evidence from an LMIC context and provides meaningful insights into gender, health, and digital access dimensions. Below, I provide comments which, if addressed in the manuscript, could enhance its suitability for publication.

Introduction

• The section in which the study objectives are presented in the Introduction (i.e., “Accordingly, this paper contributes”) would be better positioned at the end of the Introduction to improve the logical flow and clarity of the manuscript.

• It is recommended that the authors describe more studies from low- and middle-income countries (LMICs) in the Introduction to provide a broader contextual background and better situate their research within the existing literature.

Methods

• In “Quantitative Survey”: Providing additional details on the handling of missing data

• Draw Figure 1 in a larger, bold font to make it more readable.

• In “Qualitative Interviews”: Consider providing additional clarification on the participant selection criteria and the rationale for the qualitative sample size (n = 20).

• A brief description of how qualitative and quantitative findings were integrated in practice would help demonstrate the robustness of the convergent mixed-methods approach.

Result

• The results should more explicitly connect with the study’s stated objectives, particularly regarding which socio-ecological levels (individual, interpersonal, community, or system) the findings correspond to.

• Both significant and non-significant associations should be reported systematically to avoid selective emphasis and to give a balanced view of the results.

Discussion

• The discussion effectively interprets findings through the socio-ecological framework, but the link between individual-, community-, and system-level determinants could be made more explicit. Consider adding a short integrative paragraph synthesizing how these levels interact in shaping youth GAI use.

• The contrast with HIC data is informative; however, adding a sentence on why structural and cultural differences (e.g., stigma, infrastructure, digital literacy) may explain higher uptake in Pakistan would deepen the analysis.

• The conclusion is concise and well articulated. You might consider ending with a more forward-looking statement—e.g., how GAI could be ethically integrated into LMIC health systems or monitored within national digital health agendas.

Reviewer #3: The study’s methodological transparency and ethical conduct are highly commendable.

However work the following:

Expand the description of instruments and analytic frameworks within the main text.

Clarify sampling strategy limitations and justify analytical decisions.

Ensure that all analytic scripts and supporting documentation are included in OSF, with explicit links/references.

Add detail about data sharing and privacy for both survey and interview components.

The implications and conclusion are clear. Highlight future directions for addressing sampling and measurement gaps, and potentially new methodologies for reaching less-connected youth populations as digital access expands.

**Do you want your identity to be public for this peer review?** For information about this choice, including consent withdrawal, please see our Privacy Policy .

Reviewer #1: No

Reviewer #2: No

Reviewer #3: No

**Figure resubmission:** While revising your submission, we strongly recommend that you use PLOS’s NAAS tool (https://ngplosjournals.pagemajik.ai/artanalysis) to test your figure files. NAAS can convert your figure files to the TIFF file type and meet basic requirements (such as print size, resolution), or provide you with a report on issues that do not meet our requirements and that NAAS cannot fix.

**Reproducibility:** To enhance the reproducibility of your results, we recommend that authors of applicable studies deposit laboratory protocols in protocols.io, where a protocol can be assigned its own identifier (DOI) such that it can be cited independently in the future. Additionally, PLOS ONE offers an option to publish peer-reviewed clinical study protocols. Read more information on sharing protocols at https://plos.org/protocols?utm_medium=editorial-email&utm_source=authorletters&utm_campaign=protocols To enhance the reproducibility of your results, we recommend that authors of applicable studies deposit laboratory protocols in protocols.io, where a protocol can be assigned its own identifier (DOI) such that it can be cited independently in the future. Additionally, PLOS ONE offers an option to publish peer-reviewed clinical study protocols. Read more information on sharing protocols at https://plos.org/protocols?utm_medium=editorial-email&utm_source=authorletters&utm_campaign=protocols

---

## [Decision Letter · Decision Letter 1]

19 Mar 2026

Response to Reviewers '. This file does not need to include responses to any formatting updates and technical items listed in the 'Journal Requirements' section below.* A marked-up copy of your manuscript that highlights changes made to the original version. You should upload this as a separate file labeled 'Revised Manuscript with Track Changes '.* An unmarked version of your revised paper without tracked changes. You should upload this as a separate file labeled 'Manuscript '. If you would like to make changes to your financial disclosure, competing interests statement, or data availability statement, please make these updates within the submission form at the time of resubmission. Guidelines for resubmitting your figure files are available below the reviewer comments at the end of this letter. We look forward to receiving your revised manuscript. Kind regards, Harry HochheiserSection EditorPLOS Digital Health Harry HochheiserSection EditorPLOS Digital Health Leo Anthony CeliEditor-in-ChiefPLOS Digital Healthorcid.org/0000-0001-6712-6626  **Journal Requirements:** If the reviewer comments include a recommendation to cite specific previously published works, please review and evaluate these publications to determine whether they are relevant and should be cited. There is no requirement to cite these works unless the editor has indicated otherwise.  **Additional Editor Comments (if provided):** Thanks for this revision. This paper is much improved. However, Reviewer 1 has some minor comments regarding the discussion. As addressing these comments should be relatively straightforward and will likely strengthen the paper, I suggest that you review these comments and address them in a revised manuscript.**Reviewers' Comments:** Reviewer's Responses to Questions

**Comments to the Author**

Reviewer #1: (No Response)

Reviewer #3: All comments have been addressed

publication criteria ? Is the manuscript technically sound, and do the data support the conclusions? The manuscript must describe methodologically and ethically rigorous research with conclusions that are appropriately drawn based on the data presented.

Reviewer #1: (No Response)

Reviewer #3: Yes

3. Has the statistical analysis been performed appropriately and rigorously?

Reviewer #1: (No Response)

Reviewer #3: Yes

4. Have the authors made all data underlying the findings in their manuscript fully available (please refer to the Data Availability Statement at the start of the manuscript PDF file)?

Reviewer #1: (No Response)

Reviewer #3: Yes

5. Is the manuscript presented in an intelligible fashion and written in standard English?

Reviewer #1: (No Response)

Reviewer #3: Yes

Reviewer #1: you should strengthen the discussion by explicitly addressing:

How digital literacy, language, and connectivity may shape both access and health-seeking behavior.

The potential for systematic overestimation of GAI adoption.

The role of algorithmic exposure and platform familiarity in driving reported use.

Whether this population represents early adopters or structurally privileged subgroups.

The discussion would benefit from framing this cohort as a “leading edge” or innovation adoption group rather than a representative youth population.

There is something fascinating hiding inside this paper that deserves to be pushed even further. It is not just about AI use. It is about how health systems fail and how young people quietly build shadow infrastructures of care when institutions do not meet their needs. That angle, if sharpened, could turn this from a descriptive study into a foundational piece in precision public health and digital resilience research.

Reviewer #3: (No Response)

**Do you want your identity to be public for this peer review?** For information about this choice, including consent withdrawal, please see our Privacy Policy .

Reviewer #1: None

Reviewer #3: No

**Figure resubmission:** While revising your submission, we strongly recommend that you use PLOS’s NAAS tool (https://ngplosjournals.pagemajik.ai/artanalysis) to test your figure files. NAAS can convert your figure files to the TIFF file type and meet basic requirements (such as print size, resolution), or provide you with a report on issues that do not meet our requirements and that NAAS cannot fix.

**Reproducibility:** To enhance the reproducibility of your results, we recommend that authors of applicable studies deposit laboratory protocols in protocols.io, where a protocol can be assigned its own identifier (DOI) such that it can be cited independently in the future. Additionally, PLOS ONE offers an option to publish peer-reviewed clinical study protocols. Read more information on sharing protocols at https://plos.org/protocols?utm_medium=editorial-email&utm_source=authorletters&utm_campaign=protocols To enhance the reproducibility of your results, we recommend that authors of applicable studies deposit laboratory protocols in protocols.io, where a protocol can be assigned its own identifier (DOI) such that it can be cited independently in the future. Additionally, PLOS ONE offers an option to publish peer-reviewed clinical study protocols. Read more information on sharing protocols at https://plos.org/protocols?utm_medium=editorial-email&utm_source=authorletters&utm_campaign=protocols

---

## [Editor Report · Decision Letter 2]

20 Mar 2026

Use of generative AI for health among urban youth in Pakistan: A mixed-methods study

PDIG-D-25-00791R2

Dear Mashhood,

We are pleased to inform you that your manuscript 'Use of generative AI for health among urban youth in Pakistan: A mixed-methods study' has been provisionally accepted for publication in PLOS Digital Health.

Best regards,

Harry Hochheiser

Section Editor

PLOS Digital Health